# A prefrontal-thalamic circuit encodes social information for social recognition

Zihao Chen[1,7], Yechao Han [1,7], Zheng Ma [1], Xinnian Wang[1], Surui Xu[1], Yong Tang [1], Alexei L. Vyssotski [2], Bailu Si [3,4] & Yang Zhan [1,5,6] ✉

Social recognition encompasses encoding social information and distinguishing unfamiliar from familiar individuals to form social relationships. Although the medial prefrontal cortex (mPFC) is known to play a role in social behavior, how identity information is processed and by which route it is communicated in the brain remains unclear. Here we report that a ventral midline thalamic area, nucleus reuniens (Re) that has reciprocal connections with the mPFC, is critical for social recognition in male mice. In vivo single-unit recordings and decoding analysis reveal that neural populations in both mPFC and Re represent different social stimuli, however, mPFC coding capacity is stronger. We demonstrate that chemogenetic inhibitions of Re impair the mPFC-Re neural synchronization and the mPFC social coding. Projection pathway-specific inhibitions by optogenetics reveal that the reciprocal connectivity between the mPFC and the Re is necessary for social recognition. These results reveal an mPFC-thalamic circuit for social information processing.

Knowledge about familiar and unfamiliar individuals is important in directing social interactions. Failure to recognize a novel social stimulus as opposed to familiar ones underlies the core abnormalities associated with certain types of social dysfunctions[1–7]. The medial prefrontal cortex (mPFC) has been implicated in social behavior and top-down modulation of social processes[8–15]. However, little is known about how mPFC neurons encode different social stimuli and through which pathways social information might be communicated to downstream brain areas.

The thalamic nucleus reuniens (Re) has dense connections with the mPFC in a reciprocal manner[16–19]. Re is thought to facilitate communications in the prefrontal-thalamo-hippocampal route, posited to be a node linking the mPFC and the hippocampus (HPC)[20–24]. Encoding of social information necessitates the processing of multiple complex multisensory and internal cues that form an integrative representation of both acquaintance and stranger conspecifics. Given the prominent roles of the mPFC in social processing and the anatomical connectivity between the mPFC and the Re, here we test the possibility that social representation by the mPFC may utilize the pathway involving Re, and how Re may function in coordination with the mPFC to contribute to social behavior.

Prior studies have explored output pathways of the mPFC, including the amygdala[25,26] and nucleus accumbens[27]; however, despite strong anatomical connections, the mPFC-Re pathway has not been investigated. Here, we performed a combination of in vivo single-unit and multi-site electrophysiology, neural circuit manipulation, and computational analysis to explore the role of the mPFC-Re pathway in social behavior. Our results show that mPFC and Re neurons display prominent responses to social stimuli, but the response patterns in the two areas are distinct. By chemogenetic manipulations of the Re neural

[1]The Brain Cognition and Brain Disease Institute, Shenzhen Institute of Advanced Technology, Chinese Academy of Sciences, Shenzhen, China. [2]Institute of Neuroinformatics, University of Zurich and Swiss Federal Institute of Technology (ETH), Zurich, Switzerland. [3]School of Systems Science, Beijing Normal University, Beijing, China. [4]Chinese Institute for Brain Research, Beijing, China. [5]CAS Key Laboratory of Brain Connectome and Manipulation, Shenzhen Institute of Advanced Technology, Chinese Academy of Sciences, Shenzhen, China. [6]Shenzhen-Hong Kong Institute of Brain Science, Shenzhen Institute of Advanced Technology, Chinese Academy of Sciences, Shenzhen, China. [7]These authors contributed equally: Zihao Chen, Yechao Han. ✉e-mail: yang.zhan@siat.ac.cn

activity, we found that Re supports the neural synchrony between the mPFC and the Re. Next, we perturbed Re neural activity using optogenetics during the selective phase of social encoding or social recognition and reveal a crucial role of Re in social recognition but not in non-social recognition. In particular, bidirectional pathway-specific manipulations indicate that the reciprocal mPFC-Re connections are necessary for social recognition. Finally, we measured the mPFC and Re coding capacity by information-theoretic and population decoding methods. We show that chemogenetic inhibition of Re impairs mPFC social coding. Together, these results show that the mPFC-Re pathway is critical for social coding and social recognition.

## Results

### mPFC and Re responses to different social stimuli

We recorded single-unit activities in freely behaving mice during sociability and social novelty phases in the three-chamber social interaction assay[28,29] (Fig. S1a). To assess the neural response to social stimuli of familiar and novel identities, we analyzed well-isolated spikes (Fig. S2) aligned at the beginning of the investigation when the mice approached empty cup (E) and social stimulus (S) during the sociability phase, and when they approached familiar social stimulus (S') and novel social stimulus (N) during the subsequent social novelty phase (Fig. 1a). As the mice approached different stimuli and performed the investigations on them, the speeds or accelerations were not different (Fig. S1b, c), reflecting unbiased exploratory motions toward different stimuli.

In the mPFC, about 11% of neurons showed enhanced responses with increased firing and about 23% showed suppressed responses with decreased firing (90 vs. 188 of 827 neurons; Fig. 1b) when we compared the pre-stimulus firing rates and during-stimulus firing rates across all investigation bouts. In the Re, the proportions for the two types of responses were about 22% and 6% (81 vs. 22 of 365 neurons; Fig. 1c). Therefore, the proportions of the enhanced and suppressed neurons in the two areas were different. To understand the profiles of the firing rates in response to different stimuli, we normalized and averaged the firing rates of enhanced neurons within each stimulus type. We observed enhanced firing when the mice approached and investigated the four stimuli (Fig. 1d, e). The enhancement returned to the baseline level when the spike data were realigned at the end of the investigation when the mice retreated from the stimuli (Fig. 1d, e). We observed similar response profiles for the suppressed neurons (Fig. S3a, b). The firing profiles revealed a neural encoding process coinciding with the entire investigation. These data indicate that the proportions of social information encoding neurons in the mPFC and the Re were distinct.

Next, we compared the neural responses between different stimuli. In the mPFC, stimuli S, S', and N elicited stronger responses than stimulus E in the enhanced neurons (Fig. 1f). Suppressed neurons also showed similar differences between the stimuli (Fig. S3c). To examine the neural response between different stimuli at the population level, we measured the population response for each stimulus based on Mahalanobis distance[30,31] using population vectors composed of simultaneously recorded neurons. Population responses to S, S', and N were stronger than those of the shuffled data except for the population response to E (Fig. 1g). In the Re, neural responses to N and S were stronger than the response to E in the enhanced neurons. Furthermore, the response to S was stronger than the response to S' (Fig. 1h). For the suppressed neurons, the response to S was stronger than the response to E (Fig. S3d). The population responses to all stimuli were stronger than the population responses of the shuffled data (Fig. 1i). Therefore, mPFC and Re neural activities showed prominent responses to social stimuli.

### Re mediates mPFC-Re neural synchronization

To understand the interactions between the mPFC and the Re, we recorded the local field potentials (LFPs) from the two areas

simultaneously (Fig. 2a). Neural synchronization reflects how the oscillatory components communicate between different brain areas and may represent an information processing mechanism for the mPFC-Re network during social recognition. Reduced mPFC functional connectivity with subcortical areas has been reported and is associated with altered social behavior[32,33]. To investigate the neural synchronization, we examined the time-dependent coherence of the LFPs aligned at the start of the investigation (Fig. 2b). Gamma band (30–80 Hz) coherence increased during the investigation of S, S', and N (Fig. 2c). Beta band coherence also showed an increase for N (Fig. 2c). We did not observe coherence changes for E (Fig. 2c). These results indicate that social investigation is accompanied by increased neural synchronization between the mPFC and the Re.

To understand how Re may contribute to the neural synchronizations between the mPFC and the Re, we employed a combination of multi-area electrophysiology and chemogenetic designer receptors exclusively activated by designer drugs (DREADD) approach. We recorded the LFPs from the two areas and simultaneously inhibited the Re by the designer drug clozapine N-oxide (CNO) in mice injected AAV-hSyn-hM4D-Cherry in the Re (Fig. 2d, e, Fig. S4a, b). We injected AAV-hSyn-mCherry as control experiments. Because Re is thought to mediate the communications between the mPFC and the dorsal hippocampus (dHPC)[23,34], we also recorded from the dHPC. We confirmed that CNO effectively reduced neuronal firing in the Re in hM4D-expressing mice (Fig. S4c). We then examined how inhibition of Re could contribute to neural synchronization. Before the injection of the CNO, LFPs were recorded in the three-chamber environment as a baseline. After the CNO took effect, the LFPs were recorded again (Fig. 2f, g). We normalized the coherence using the baseline and found that mPFC-Re coherence was reduced in the CNO treatment group compared to the vehicle (VEH) group in gamma band in hM4D-expressing mice (Fig. 2i). In mCherry-expressing mice, no difference was found before the two treatment groups (Fig. 2i). In contrast to mPFC-Re coherence, the CNO did not change Re-dHPC or mPFC-dHPC coherence results (Fig. S5a, b). Gamma band power displayed reduction when the Re was inhibited in the hM4D group (Fig. S5c–e). These data demonstrate that normal Re functioning is necessary for the neural synchronization between the mPFC and the Re.

### Re supports mPFC social representation

To investigate how Re can contribute to the mPFC representation of social information, we recorded the single-unit activity in the mPFC while simultaneously inhibiting the Re using DREADD (Fig. 3a). Interestingly, compared to the VEH treatment group, the proportions of enhanced and suppressed units were altered by the CNO treatment (CNO group, 79 vs. 72 of 430 neurons; VEH group, 103 vs. 143 of 581 neurons; Fig. 3b, c), indicating that inhibition of the neural activities in the Re can affect the responsive subpopulations in the mPFC. The firing profiles showed that enhanced neurons responded to the stimuli during the investigations (Fig. 3d, e, Fig. S6a, b). In the CNO treatment group, only the neural responses of S and S' differed from the response of E (Fig. 3f). In contrast, in the VEH treatment group, the neural responses to S, S', and N were higher than the response to E (Fig. 3g), similar to the mPFC neural responses in Fig. 1f. Suppressed neurons also displayed altered neural responses of different stimuli in the CNO group compared to the neural responses in the VEH group (Fig. S6c, d). Inhibition of Re therefore could influence the neural representations of social stimuli in the mPFC.

To understand how each neuron was selective for one stimulus compared to another, we calculated the selectivity index for all the neurons in the two treatment groups. In the VEH treatment group, there existed subpopulations showing very strong preferences for S in the sociability phase as the selectivity index distribution showed a peak at the end corresponding to being perfectly selective for S (Fig. 3h). In the CNO treatment group, subpopulations

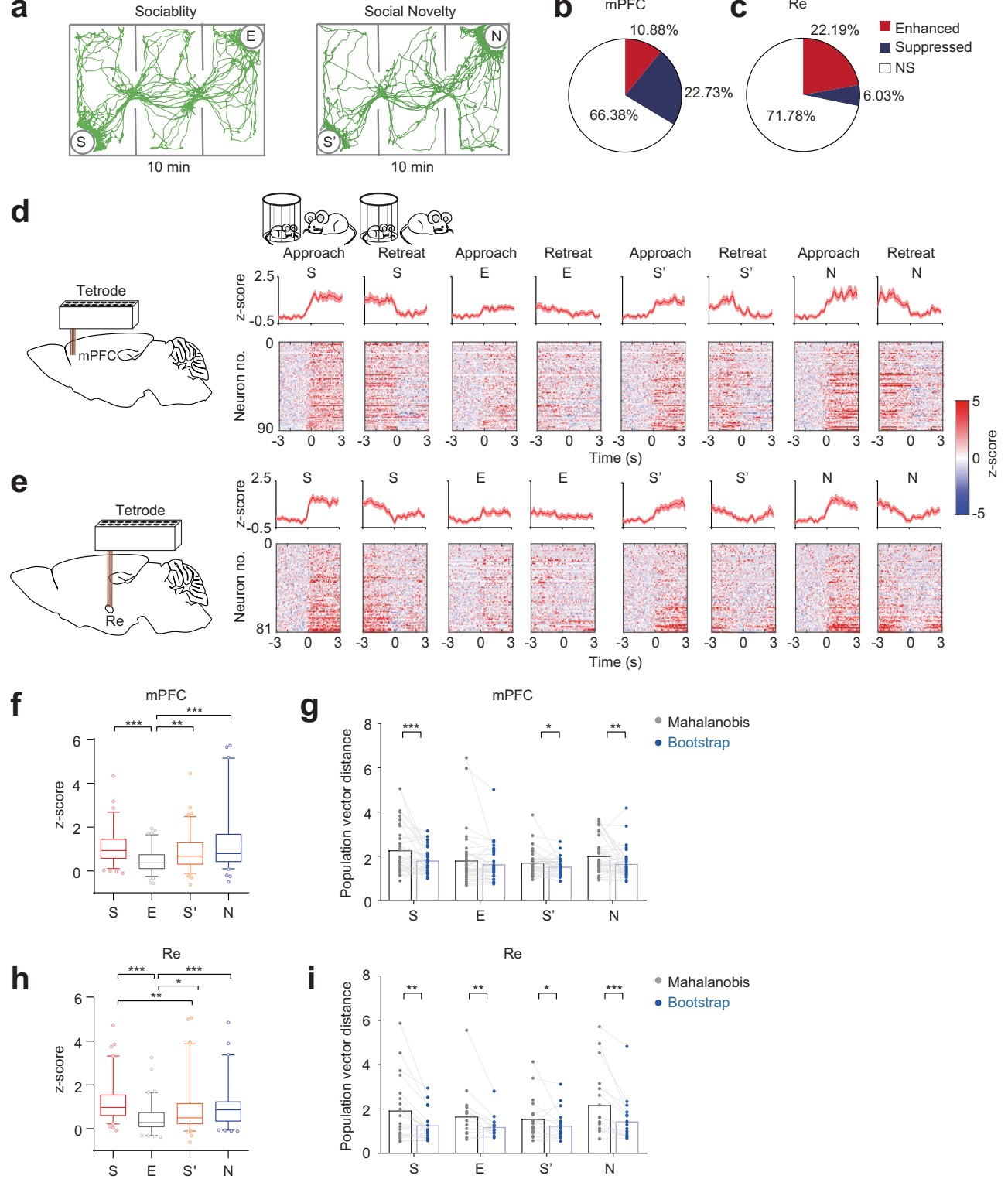

**Fig. 1 | mPFC and Re neural responses to social stimuli. a** Representative tracking traces in three-chamber social interaction test. **b**, **c** Proportion of enhanced, suppressed and non-significant neurons to the stimuli in the mPFC and the Re. Chi-square test, two-sided, $\chi^2 = 63.77$, $P < 0.0001$. mPFC: $n = 827$ neurons. Re: $n = 365$ neurons. NS, non-significant. Response profiles of enhanced neurons to S, E, S' and N aligned at the beginning (approach) and the end (retreat) of the investigation for mPFC (**d**) and Re (**e**). Response profiles were calculated as z-scores within stimulus type. Neurons were sorted by the magnitude (lowest to highest) of the average response across all investigation bouts. Line and shaded areas are mean ± s.e.m. **f** mPFC responses for each stimulus aligned at the beginning of the investigation. Kruskal-Wallis test with Dunn's multiple comparison correction. $n = 90$ neurons.

$H(3) = 40.94$, $P < 0.0001$. Box plots showing median, 25%–75% percentile, 5%–95% range and outliers. **g** Population vector response between original and shuffled data using Mahalanobis distance for different social stimuli. The shuffling was performed using the bootstrap method. Wilcoxon signed rank test, two-sided. S, $n = 38$ sessions, $P = 0.003$; E, $n = 37$ sessions; S', $n = 38$ sessions, $P = 0.0206$; N, $n = 37$ sessions, $P = 0.0035$. **h**, **i** The same as (**f**, **g**) but for Re. **h** $n = 81$ neurons. $H(3) = 44.48$, $P < 0.0001$. **i** S, $n = 21$ sessions, $P = 0.0016$; E, $n = 18$ sessions, $P = 0.0077$; S', $n = 20$ sessions, $P = 0.04$; N, $n = 21$ sessions, $P < 0.0001$. *$P < 0.05$, **$P < 0.01$, ***$P < 0.001$. Details of the statistical information are provided in Supplementary Data 1.

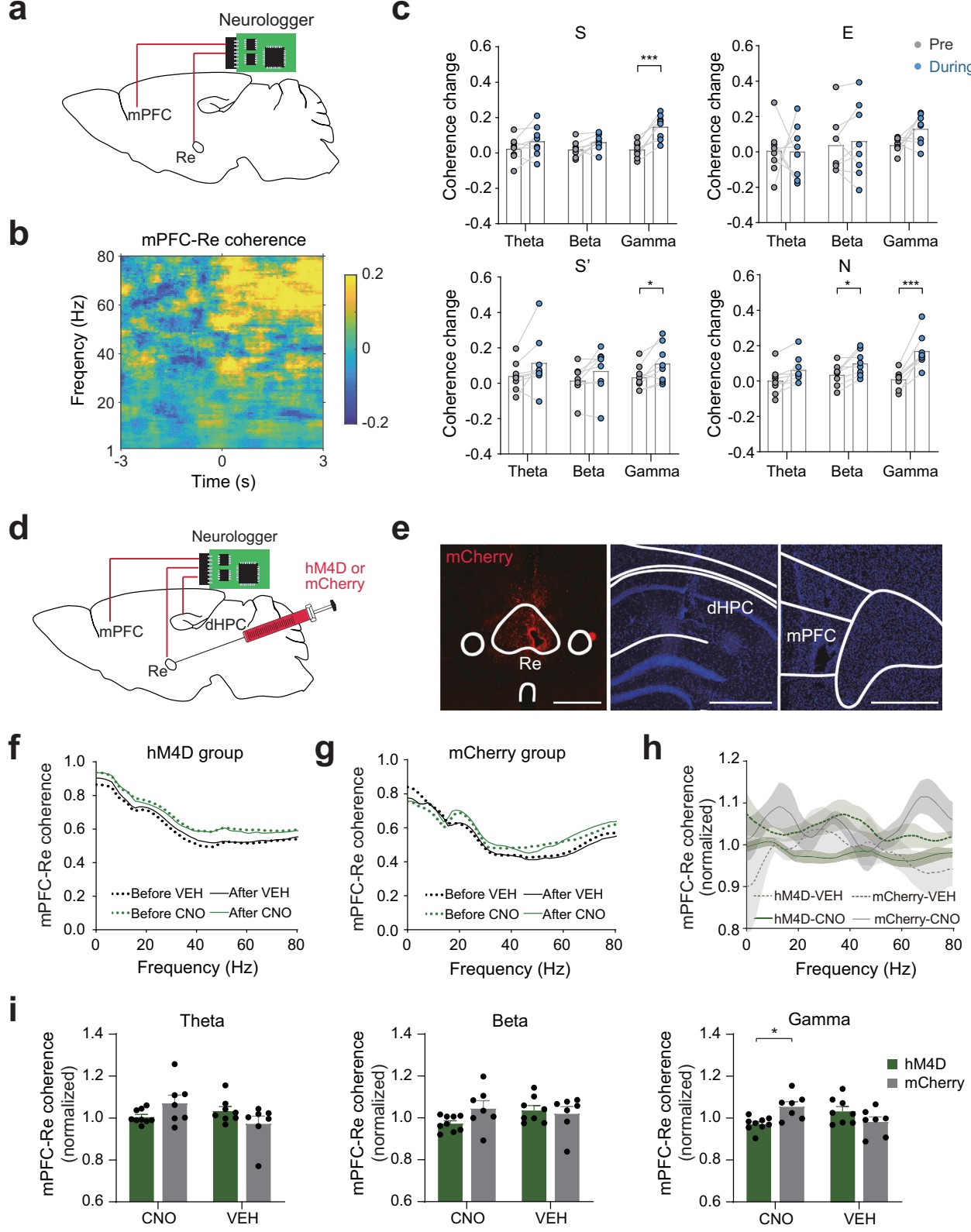

showed ambiguous selectivity with the index distribution having a unimodal peak centered around zero (Fig. 3h). The S vs. E selectivity indexes between the two treatment groups had different distributions. In the social novelty phase, the distributions of N vs. S' index between the two treatment groups were also different (Fig. 3h). Therefore, inhibition of the Re affects the neuronal selectivity for social stimuli in the mPFC.

## Re is required for social recognition behavior

To investigate how Re can contribute to social behavior, we employed the optogenetic approach. AAV- hSyn-eNpHR3.0-GFP or AAV-hSyn-GFP was injected in the Re, and optic fibers were implanted above it (Fig. 4a, Fig. S7a). We confirmed that yellow light illumination effectively reduced the neuronal firing in the Re in the eNpHR mice (Fig. S7b). We first switched on the yellow light during both phases of

**Fig. 2 | Re mediates mPFC-Re neural synchronization. a** Schematic showing the LFP recordings in the mPFC and the Re. **b** Time-dependent coherence between the mPFC and the Re. Coherence was normalized using the pre-stimulus period data across the frequency. **c** Coherence changes from the pre-stimulus period for S, E, S' and N at different frequency bands. Paired $t$ test, two-sided. $n = 8$ mice. S: gamma, $t(7) = 6.038$, $P = 0.0005$; S': gamma, $t(7) = 3.263$, $P = 0.0138$; N: beta, $t(7) = 3.29$, $P = 0.0133$; gamma: $t(7) = 6.636$, $P = 0.0003$. **d** Schematic showing multi-site LFP recordings and inhibition of Re using DREADD method. **e** Brain sections showing expression of hM4D in the Re and the placement of electrodes in the mPFC, Re and dHPC. Scale bar, 500 μm. **f, g** Average mPFC-Re coherence before and after the injections of CNO or VEH in mice expressing hM4D or mCherry. **h** Normalized coherence for CNO and vehicle (VEH) treatment groups in mice expressing hM4D or mCherry. Line and shaded areas are mean ± s.e.m. **i** Coherence for theta, beta and gamma bands. Coherence was averaged within each frequency bands. Two-way ANOVA with Bonferroni correction. hM4D group, VEH: $n = 8$ mice. CNO: $n = 9$ mice. mCherry group, VEH: $n = 7$ mice. CNO: $n = 7$ mice. Gamma coherence, treatment × drug $F(1, 27) = 11.61$, $P = 0.0021$. Data in (**c, i**) are mean ± s.e.m. *$P < 0.05$; ***$P < 0.001$. Details of the statistical information are provided in Supplementary Data 1.

sociability and social novelty. In the sociability phase, we found that both the eNpHR and the GFP groups spent more time exploring S than E (Fig. 4b), indicating that sociability was unaffected. In the social novelty phase, the GFP group exhibited a preference for spending more time with N than S' (Fig. 4b); in contrast, the eNpHR group did not show a difference in time spent between N and S' (Fig. 4b). In addition, in mice expressing inhibitory DREADD in the Re the time between N and S' was not different under the CNO treatment (Fig. S8a, b). These results indicate that inhibition of the Re impaired social recognition between the familiar and novel mice.

We next inhibited the Re during either the sociability or the novelty phase to understand how phase-selective inhibition can affect social recognition. We first inhibited the Re during the sociability phase. A recognition impairment between N and S' was found in the social novelty phase (Fig. 4c), although the light was not switched on during this phase. We then inhibited the Re during the social novelty phase in a separate cohort of mice. Photoinhibition also impaired the social recognition between N and S' (Fig. 4d). These results demonstrate that Re is necessary for social recognition.

We then examined social recognition in the five-trial social interaction assay. In this assay, a stimulus mouse was presented for four successive trials, and on the fifth trial, a novel stimulus mouse was introduced. In the control group, the investigation time gradually decreased when the first stimulus mouse was repeatedly presented and increased in the last trial when a second stimulus mouse was presented (Fig. 4e). In contrast, the investigation time in the eNpHR group did not change across the trials (Fig. 4e). These results indicate that inhibition of Re impairs social recognition in a different behavioral assay.

To rule out that inhibition of Re may affect the recognition of non-social stimuli, we performed the object recognition test. We found that both eNpHR and GFP groups spent more time with the novel object than the familiar objects (Fig. 4f), indicating that optogenetic inhibition of the Re did not affect object recognition. To test whether Re inhibition affects the recognition of olfactory cues, we performed the odor recognition test in the same three-chamber environment. We presented one odor from the soiled bedding of a home cage during the first phase and introduced a second odor from another home cage during the subsequent recognition phase. We found that optogenetic inhibition of the Re did not affect the preference for a novel odor (Fig. 4g). Therefore, Re is necessary for social recognition and its inhibition does not affect object or social odor recognition.

### Bidirectional modulation of social recognition in the mPFC-Re circuit

The mPFC and the Re have bidirectional synaptic connections[35–37]. To understand the role of axonal projections in either mPFC-Re or Re-mPFC directions for social behavior, we performed optogenetic inhibitions at the axon terminals. First, we injected AAV-hSyn-eNpHR3.0-GFP or AAV-hSyn-GFP into the bilateral sides of the mPFC and implanted the optic fiber above the Re (Fig. 5a). Inhibition was performed in separate phases of the three-chamber social interaction test in separate cohorts of mice. We found that inhibition of the projections in the Re during sociability phase impaired social recognition (Fig. 5b). When the inhibition was performed during the social novelty phase, we did not find a group by stimulus interaction but the N vs. S' preference indexes were different between the two groups (Fig. 5c). The S vs. E social preference in the sociability phase was not affected. Next, we injected the viruses in the Re and implanted the optic fibers in the mPFC bilaterally (Fig. 5d). We also found that inhibition of the thalamic projections in the mPFC during either phase impaired social recognition (Fig. 5e, f). Thus, neural transmissions in both directions along the mPFC-Re pathway are necessary for social recognition.

### Inhibition of Re influences mPFC social coding

To further understand how mPFC and Re can represent different social stimuli, we developed an information-theoretic measurement to characterize the social information. Social information was computed by the mutual entropy[38] between the firing response of the population vectors and the occurrence of investigating different stimuli (Fig. 6a). The information analysis quantifies the reduction of uncertainty about the social stimuli that can be gained from the observation of the neural response[39]. Using binned population vectors and information measurements, we found that mPFC had higher information than Re (Fig. 6b). We next performed machine-learning classifications to investigate the decoding performance in the two brain areas. We trained a naïve Bayesian classifier with binned population vectors to decode the four stimuli. We found that the decoding performance increased with the population dimension (Fig. 6c). At higher dimensions, the classification accuracy in the mPFC was higher than in the Re (Fig. 6c). Therefore, population-level information and decoding analyses reveal that social coding capability in the mPFC is stronger than in the Re.

To understand how Re can contribute to the mPFC social coding, we performed the information analysis of the population vectors in the mPFC when the Re was chemogenetically inhibited by the inhibitory DREADD. We found that CNO treatment reduced social information (Fig. 7a), demonstrating that Re is necessary for social information coding in the mPFC. Next, we performed the single-trial decoding analysis. Before the decoding, we applied spatial filtering to the population vectors to reduce the influence of noise (Fig. 7b). Spatial filters transform the high-dimensional population vectors with supervised learning to a latent representation. Then the latent representation was subject to the decoding analysis. We found that CNO treatment reduced the decoding accuracy for the two different social stimuli of S and N (Fig. 7c). As a control analysis for the single-trial decoding, we shuffled the behavioral bouts and the corresponding firing responses and performed the decoding analysis. We did not find different decoding accuracy between CNO and VEH groups (Fig. 7c). Therefore, information and decoding analyses demonstrate that Re supports the mPFC coding capability for discrimination of different social stimuli.

### Discussion

This study identifies a prefrontal-thalamic circuit essential for information coding about social targets. Here, using in vivo single-unit recordings in freely-moving mice and decoding analysis, we show

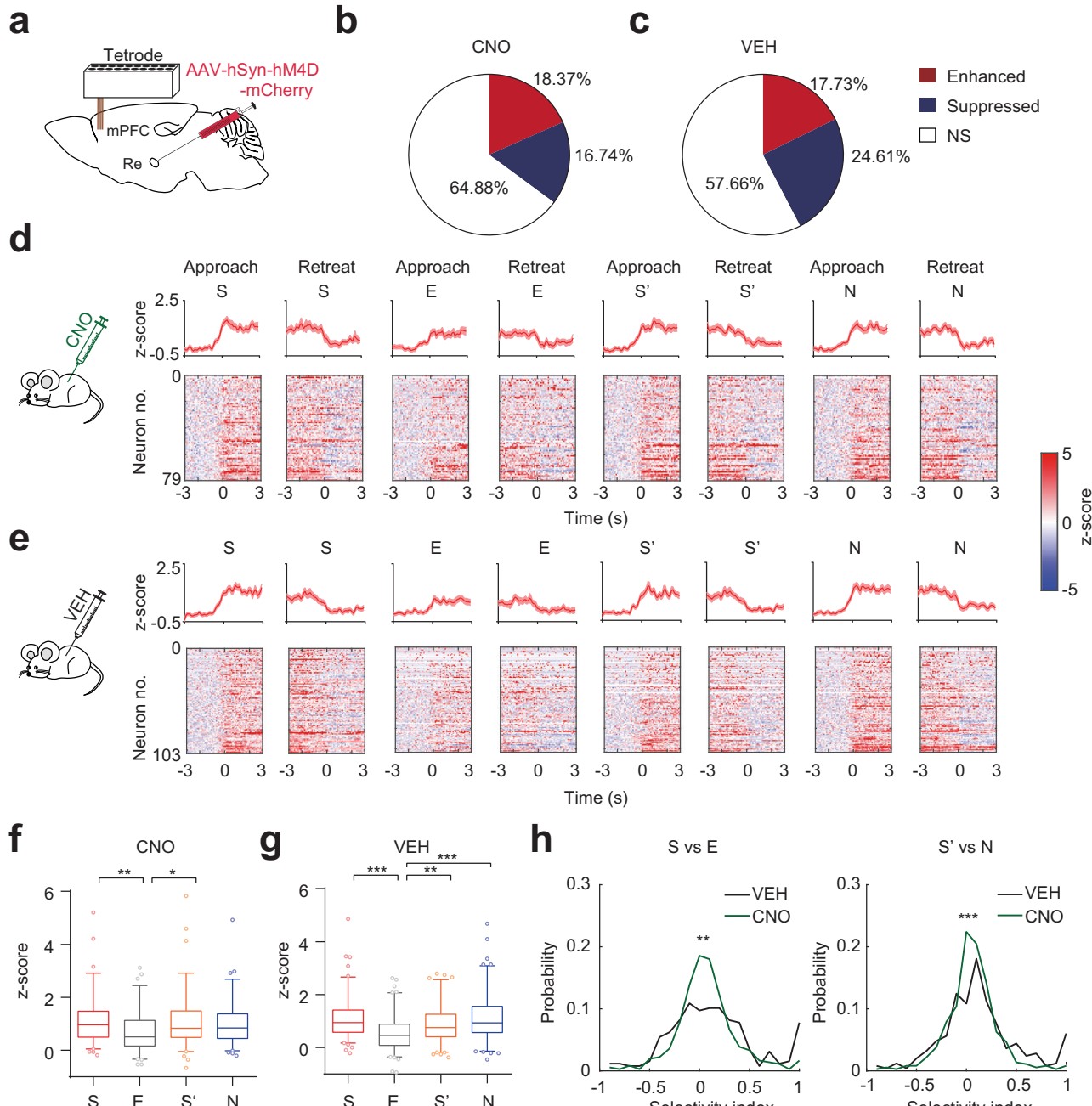

**Fig. 3 | Altered mPFC social responses when the Re is inhibited. a** Schematic showing the expression of hM4D in the Re and the tetrode recordings in the mPFC. **b**, **c** Proportion of enhanced, suppressed, and non-significant neurons to the stimuli in the mPFC under the treatment of CNO or VEH. Re. Chi-square test, two-sided, $\chi^2=9.375$, $P=0.0092$. CNO: $n=430$ neurons. VEH: $n=581$ neurons. NS, non-significant. **d**, **e** Response profiles of enhanced neurons to S, E, S' and N aligned at the beginning (approach) and the end (retreat) of the investigation under the treatment of CNO or VEH. Line and shaded areas are mean ± s.e.m. **f**, **g** mPFC responses for each stimulus aligned at the beginning of the investigation in CNO

and VEH groups. Kruskal-Wallis test with Dunn's multiple comparison correction. **f** CNO: $n=79$ neurons. $H(3)=13.33$, $P=0.004$. **g** VEH: $n=103$ neurons. $H(3)=33.34$, $P<0.0001$. Box plots showing median, 25%–75% percentile, 5–95% range and outliers. **h** Distribution of the selectivity index during sociality (S vs. E) and social novelty phase (N vs. S'). The index was calculated as $(R_S-R_E)/(R_S+R_E)$ for the sociability phase and as $(R_N-R_{S'})/(R_N+R_{S'})$ for the social novelty phase. R was the firing response. Kolmogorov–Smirnov test, two-sided, S vs. E: $D=0.1536$, $P=0.0017$. S' vs. N: $D=0.1609$, $P=0.0009$. *$P<0.05$; **$P<0.01$; ***$P<0.001$. Details of the statistical information are provided in Supplementary Data 1.

that subsets of mPFC and Re neural populations encode information about familiar and novel identities. Previous work has shown that mPFC neurons in non-human primates exhibit increased firing and distinct neuronal patterns for novel and familiar faces[40]. mPFC contains neurons responsive to social stimuli[10,11,41,42]; however, the functional role of the Re in social behavior has not been explored. We demonstrate a critical role of the ventral midline thalamus in social behavior.

Re, as a major thalamic target interconnected with the mPFC, has been investigated regarding the specificity and generalization of memorized contexts[24,43,44]. Here we report that the Re is critical for social recognition and the neural synchronization between the mPFC and the Re. Neural synchronization is important for efficient neuronal communications[45,46] and impaired synchronization has been associated with social deficits[33]. Computational models using reciprocally connected echo-state or reservoir networks[47,48] have demonstrated

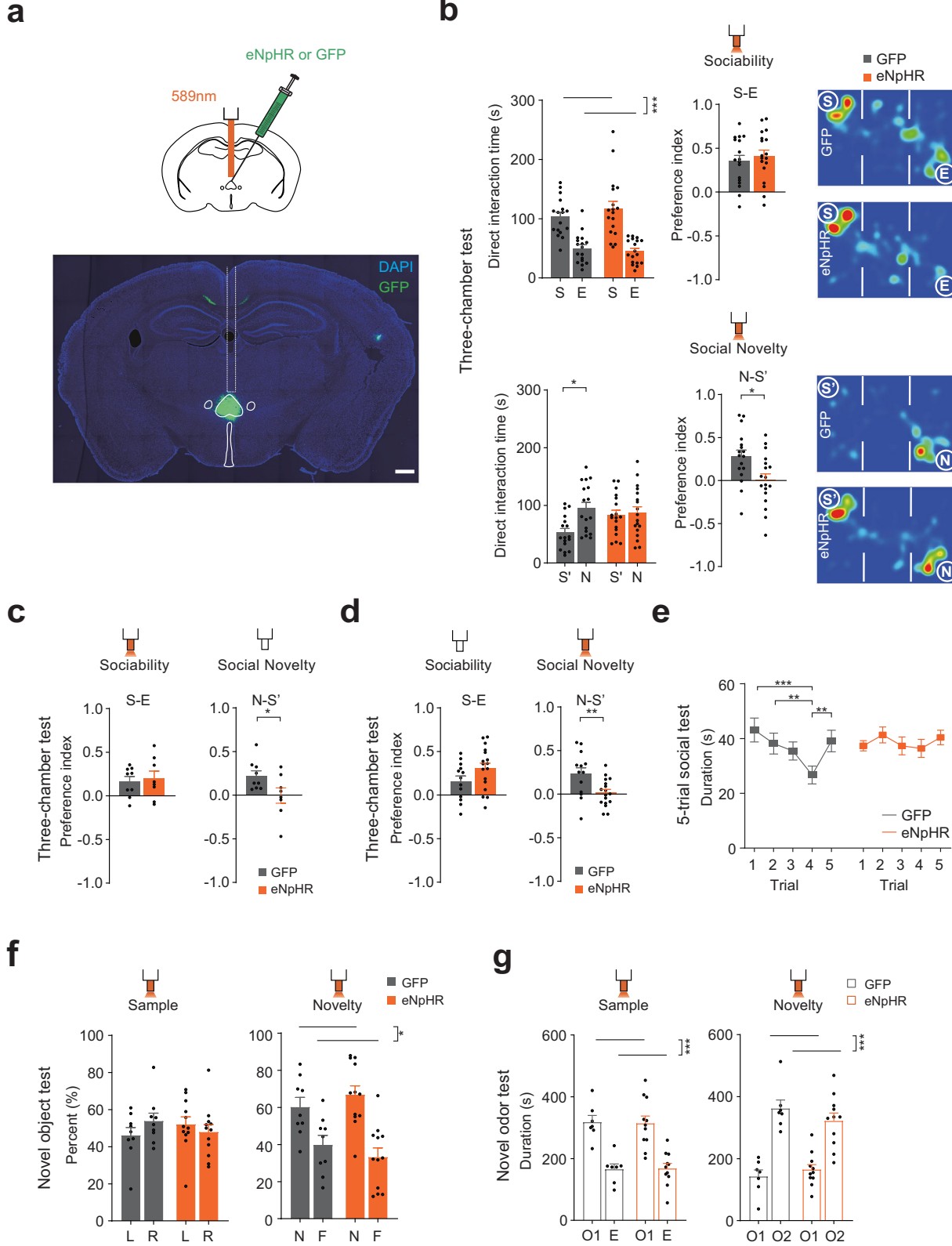

that synaptic modifications to the projection subpopulations are critical for pattern representation[49,50]. The bidirectionally connected mPFC-Re circuit may support efficient social processing to represent different social stimuli.

mPFC exerts top-down control[12,14] and the mPFC-Re pathway may constitute an important route for the transmission of targeted information. In light of divergent mPFC descending pathways in thalamic areas and other subcortical areas[25,27,51,52], interactions between the mPFC and the Re may contribute to the fulfillment of social task out of many different tasks mPFC is involved in[53]. Further studies will be needed to investigate how various afferent or efferent subpopulations operate the principle of computations. Our findings underscore the importance of mPFC in social recognition and provide insights into neural information processing and transmission along cortical-thalamic pathways.

**Fig. 4 | Re mediates social recognition. a** Optogenetic inhibition of the Re. Brain section image showing expression of eNpHR3.0 in the Re. Inset, schematic showing optogenetic inhibition of Re using yellow light. Scale bar, 500 μm. **b** Time spent investigating the different stimuli, preference index and representative heat maps of the dwell time for sociability (10 min) and social novelty (10 min) phases in the three-chamber test when the Re was optogenetically inhibited. E, empty cup. S, first social stimulus. S', the same first social stimulus that became familiar during the social novelty phase. N. novel social stimulus. Preference index during sociability phase was calculated as the difference time divided by the total time between S and E. Preference index during social novelty phase was the same between N and S'. Direct interaction time was compared using two-way ANOVA followed by Bonferroni correction. GFP: $n = 17$ mice, eNpHR: $n = 18$ mice. Sociability: stimulus $F_{(1,66)} = 56.59$, $P < 0.0001$; Social novelty: stimulus × group $F_{(1,66)} = 4.3665$, $P = 0.0405$. Preference index was compared using two-sided unpaired $t$ test. S-N index, $t_{(33)} = 2.663$, $P = 0.0119$. **c, d** Preference index after inhibition of Re during sociability or social novelty phase in the three-chamber test. Unpaired $t$ test, two-sided. **c** GFP: $n = 9$ mice, eNpHR: $n = 8$ mice. N-S' index, $t_{(15)} = 2.23$, $P = 0.0414$. **d** GFP: $n = 13$ mice, eNpHR: $n = 17$ mice. N-S' index, $t_{(28)} = 2.83$, $P = 0.0085$. **e** Investigation time in the 5-trial habituation-dishabituation test with optogenetic inhibition of the Re. Two-way repeated measures ANOVA with Bonferroni correction. GFP: $n = 9$ mice, eNpHR: $n = 11$ mice. Trial × group $F_{(4,72)} = 3.217$, $P = 0.0174$. **f** Investigation time in the object recognition test with the optogenetic inhibition of the Re. L: Left object. R: Right object. F: Familiar object. N: Novel object. Percent of time was calculated as the time spent in investigating the target object divided by the total time investigating the two objects. Two-way ANOVA with Bonferroni correction. GFP: $n = 9$ mice, eNpHR: $n = 11$ mice. Novelty: object $F_{(1,38)} = 27.6$, $P < 0.0001$. **g** Time spent in different chambers in the odor recognition test with the optogenetic inhibition of the Re. E, empty cup. O1, first odor. O2, second odor. Two-way ANOVA with Bonferroni correction. GFP: $n = 7$ mice, eNpHR: $n = 11$ mice. Sample: odor $F_{(1,32)} = 48.3$, $P < 0.0001$; Novelty: odor $F_{(1,32)} = 65.41$, $P < 0.0001$. *$P < 0.05$; **$P < 0.01$; ***$P < 0.001$. Data are mean ± s.e.m. Details of the statistical information are provided in Supplementary Data 1.

## Materials and methods

### Animals
Adult (over two months old) male C57BL/6J mice were used as test mice. Juvenile (3–5 weeks old) male C57BL/6J mice were used as stimulus mice in the three-chamber social test. All mice were group-housed under standard housing conditions (12:12-h light/dark cycle, 22–25 °C, 40–70% humidity) with food pellets and water ad libitum. Animal husbandry and experimental manipulation in this study were approved by Animal Care and Use Committees at the Shenzhen Institute of Advanced Technology (SIAT), Chinese Academy of Sciences (CAS), China.

### Three-chamber social interaction test
The social interaction test[33,54–56] was conducted in a rectangular, three-chamber apparatus (60 cm × 40 cm). The two separation walls had gaps (10 cm) in the middle to allow mice access to all the chambers. One cylinder cup with metal bars was placed in the corner of the apparatus, and an identical cup was placed in the opposite corner. In the habituation phase, the test mouse was placed in the middle of the apparatus and allowed to explore the environment freely for 5 min. In the sociability phase, an unfamiliar male juvenile with no prior contact with the test mice was placed in one of the two cups. The test mouse was allowed to explore for 10 min. In the following social novelty phase, a novel mouse was introduced to the other cage, and the test mice freely explored the environment for 10 min. The behavior was monitored by an overhead camera. The behavioral videos were analyzed by ANY-maze software (version 7.3, Stoelting Co.). Separate cohorts of mice were used when we optogenetically inhibited different phases of the three-chamber social interaction test.

### Habituation-dishabituation social memory test
Social memory test[1,57] was conducted in a rectangle box (18 cm × 16 cm) similar to the home cage. Ovariectomized female CD1 mice were used as stimulus mice. The test mouse was placed in the middle of the apparatus and allowed to habituate freely for 5 min. After the habituation, the first interaction trial started by placing one stimulus mouse in the middle of the box for one minute to allow the two mice to interact. After the first trial, the stimulus mouse was returned to the holding cage. This procedure was repeated four times with 10-min intervals in between. In the fifth trial, a novel stimulus mouse was introduced as a probe trial for the social novelty test. The duration of investigatory behavior (anogenital sniffing, body and tail sniffing, and head sniffing) was manually recorded.

### Object recognition test
The object recognition test[58] was conducted in a square box (40 cm × 40 cm). In the habituation period, the test mouse was placed in the middle of the apparatus and allowed to explore freely for five minutes. In the sample-object exposure phase, two identical objects were placed in the left and right corners of the box. The test mouse was placed in the center of the box and allowed 10 min to investigate the objects freely. After this phase, one of the objects would be replaced by a novel object and then the test mouse spent another 10 min exploring the environment. The mice in the novel object test had previously participated in the three-chamber social test. Before the novel object test, mice were given at least one week to rest in their home cages. The duration of investigation time was recorded by manually inspecting the video frames corresponding to the start and the end of the investigation.

### Odor recognition test
The odor recognition test was performed in the same box as the three-chamber social interaction test. The test mouse was allowed 5 min to habituate to the arena, including two empty cylinder cups. A pinch of sawdust bedding containing the social odor from the same home cage where the test mouse was housed was randomly put into one of the cups. The test mouse was placed in the center chamber and allowed 10 min to explore the test environment freely. After the first phase, the same amount of sawdust from another cage that housed other groups of mice and contained a novel odor was put into the other empty cup. Then the test mouse spent another 10 min exploring the environment. The total time spent in each chamber was analyzed and recorded by ANY-maze software.

### Optogenetic manipulation
Stereotaxic surgeries and virus injections were performed on a stereotaxic frame (RWD, China). Mice were anesthetized with sodium pentobarbital (80 mg/kg) and maintained with 1% isoflurane. AAV-syn-eNpHR3.0-EYP and AAV-syn-EYP were packaged locally and the titers were in the range of $2–5 × 10^{12}$ genome copies/mL. The virus was injected via a needle using a Hamilton syringe. 100 nL volume was injected controlled by a micro-injection pump. The coordinates for virus injections in the Re were anterior-posterior (AP): −1.34 mm, medial-lateral (ML): 0 mm, dorsal-ventral (DV): 4.2 mm. In the mPFC, the coordinates were AP: +1.80 mm, ML: 0.3 mm, DV: 2.20 mm. After one-week recovery period, an optic fiber coupled with a ceramic ferrule (fiber diameter 200 μm) was implanted into the Re (AP: −1.30 mm, ML: 0 mm, DV: 3.8 mm) or the mPFC (AP: +1.8 mm, ML: 0.8 mm, DV: 1.87 mm, 15° lateral midline angle) and fixated to the skull using dental cement. The ferrule was connected to the laser source (Newdoon, China) via an optic fiber sleeve. For optogenetic inhibitions during behavior, yellow light was applied continuously (589 nm, 6 mW power).

### DREADD manipulation
Viral infection was performed on five weeks old mice. AAV-hSyn-hM4Di-mCherry or control virus AAV-hSyn-mCherry was injected into

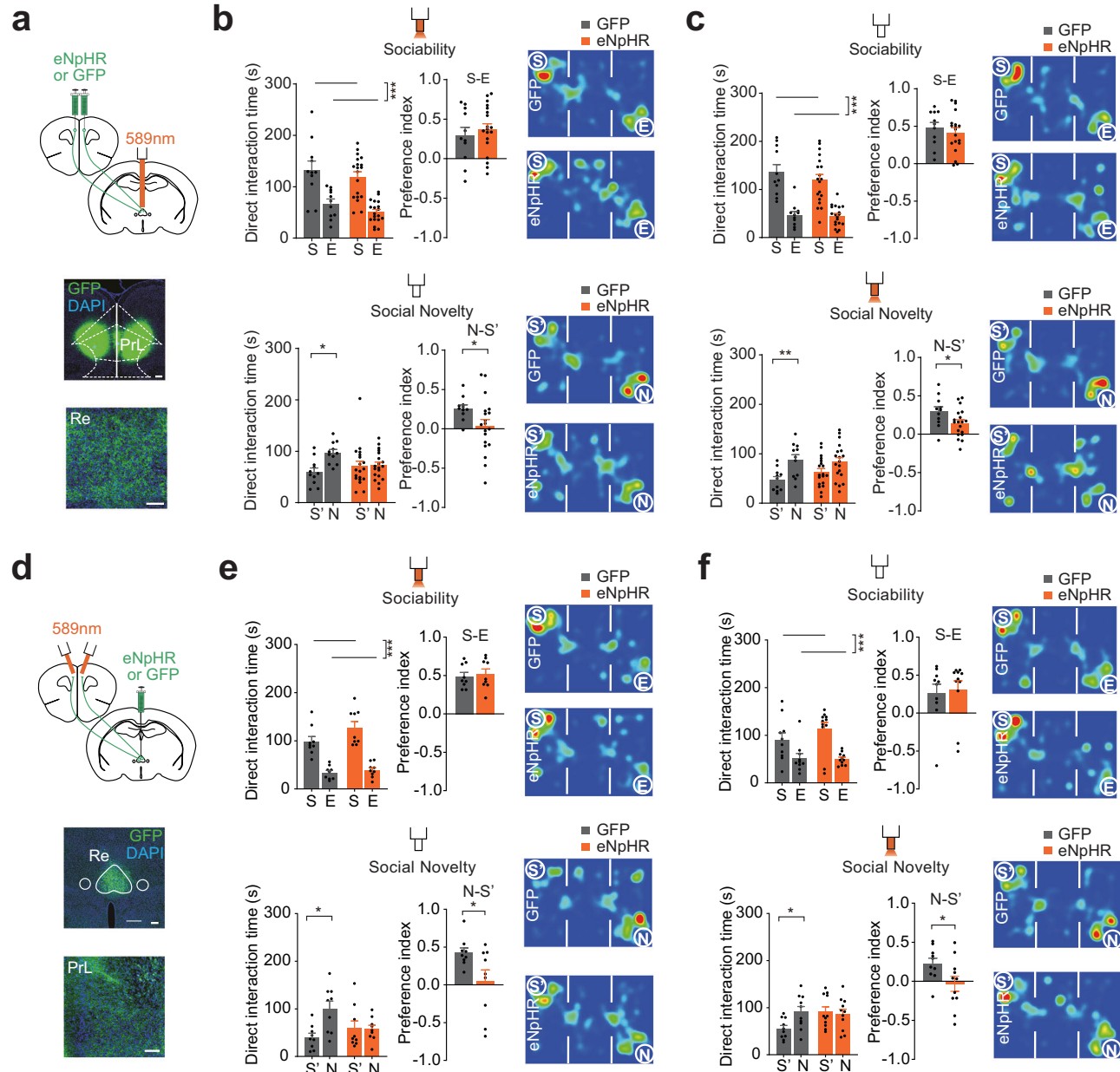

**Fig. 5 | Bidirectional inhibitions of mPFC-Re pathways impair social recognition. a** Schematic showing optogenetic inhibition of mPFC terminals in the Re using yellow light. Brain sections showing the expression of eNpHR3.0 in the bilateral mPFC and the axon terminals in the Re. Scale bar, 500 μm (top) and 100 μm (bottom). **b, c** Time spent investigating the stimuli, preference index and representative heat maps after mPFC-Re projection inhibition at different phases in the three-chamber test. **b** GFP: $n = 11$ mice, eNpHR: $n = 19$ mice. Direct interaction time was compared using two-way ANOVA followed by Bonferroni correction. Sociability: stimulus $F(1,54) = 48.9$, $P < 0.0001$; Social novelty: stimulus × group $F(1,54) = 6.142$, $P = 0.0164$. Preference index was compared using two-sided unpaired t test. N-S' index, $t(27) = 2.468$, $P = 0.0202$. **c** GFP: $n = 11$ mice, eNpHR: $n = 18$ mice. Sociability: stimulus $F(1,56) = 64.3$, $P < 0.0001$; Social novelty: stimulus × group $F(1,56) = 1.311$, $P = 0.2572$. N-S' index, $t(28) = 2.106$, $P = 0.0443$.

**d** Schematic showing optogenetic inhibition of Re terminals in the bilateral mPFC using yellow light. Brain sections showing the expression of eNpHR3.0 in the Re and the axon terminals in the mPFC. Scale bar, 500 μm (top) and 100 μm (bottom). **e, f** Time spent investigating the stimuli, preference index and representative heat maps after Re-mPFC projection inhibition at different phases in the three-chamber test. **e** GFP: $n = 9$ mice, eNpHR: $n = 9$ mice. Sociability: stimulus $F(1,32) = 79.55$, $P < 0.0001$; Social novelty: stimulus × group $F(1,32) = 6.131$, $P = 0.0188$. N-S' index, $t(16) = 2.316$, $P = 0.0342$. **f** GFP: $n = 10$ mice, eNpHR: $n = 11$ mice. Sociability: stimulus $F(1,38) = 20.79$, $P < 0.0001$; Social novelty: stimulus × group $F(1,38) = 4.149$, $P = 0.0487$. N-S' index, $t(19) = 2.182$, $P = 0.0419$. *$P < 0.05$, **$P < 0.01$, ***$P < 0.001$. Data are mean ± s.e.m. Details of the statistical information are provided in Supplementary Data 1.

the Re, and three weeks were allowed for surgery recovery and virus expression. On the test day, CNO was injected intraperitoneally (3 mg/kg) 35 min before the social interaction test.

**Local field potential electrophysiology**

Mice were anesthetized, the skull was exposed, and burr holes were drilled to implant electrodes. Nichrome wire electrodes (65 μm in diameter, A-M System, Inc) were individually implanted in the left pre-limbic area of mPFC (AP: +1.8 mm, ML: 0.5 mm, DV: 2.2 mm), Re (AP: −1.30 mm, ML: 0 mm, DV: 4.4 mm) and dHPC (AP: −1.5 mm, ML: 1.0 mm, DV: 1.5 mm). Two additional screws were placed at the contralateral frontal area and the area above the cerebellum to serve as the reference and ground, respectively. All wires were carefully inserted into a 7-pin socket, which served as the interface with the Neurologger[33]. The

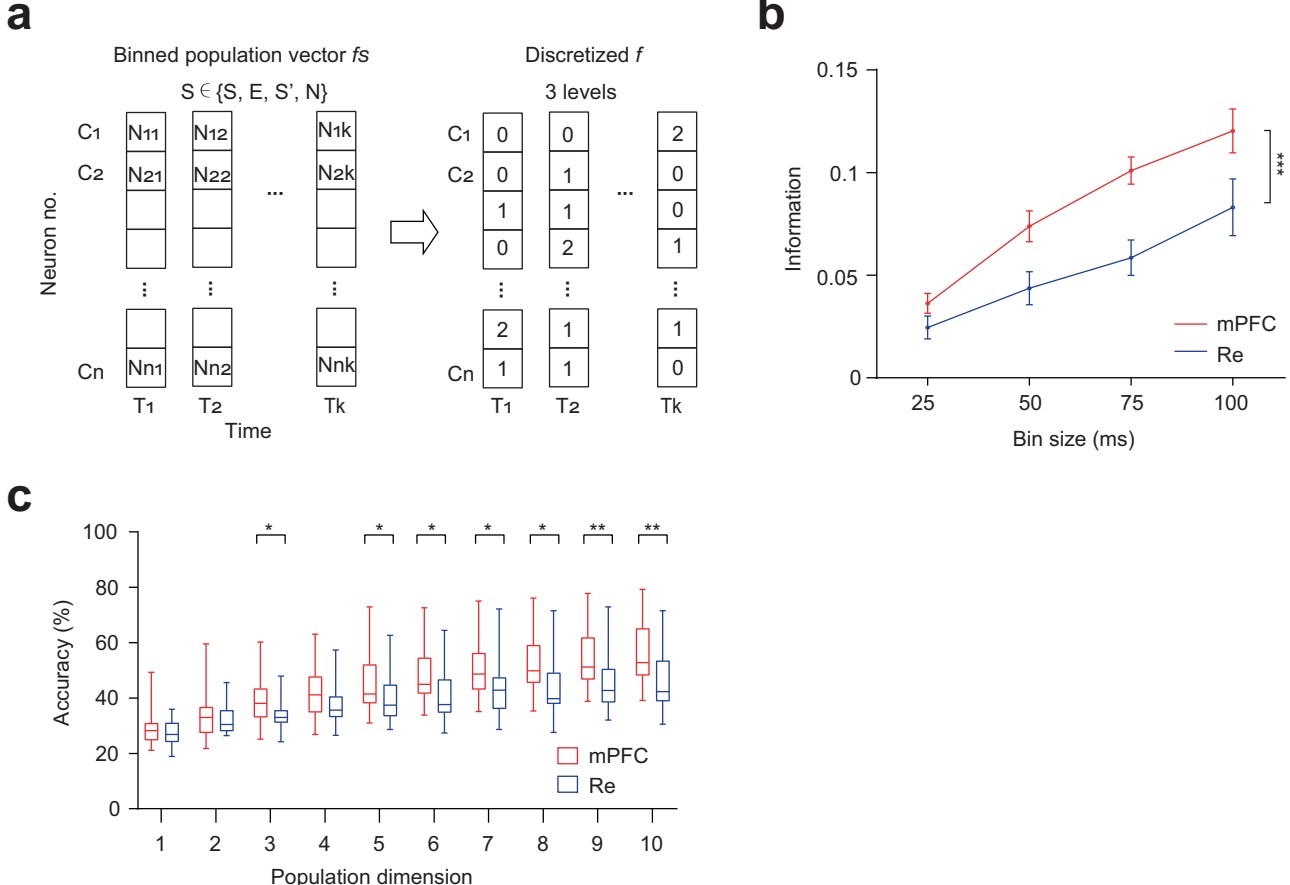

**Fig. 6 | The social coding in the mPFC is stronger than in the Re. a** Illustration of binned population vectors that were used for Shannon entropy and social information calculation. **b** Information measurements for mPFC and Re population vectors for S, E, S' and N. mPFC: $n = 39$ sessions, Re: $n = 26$ sessions. Two-way ANOVA. Brain region $F(3, 252) = 24.82$, $P < 0.0001$. Data are mean ± s.e.m. **c** Decoding accuracy for mPFC and Re population vectors with different population dimensions in the classification of S, E, S' and N. Mann–Whitney test, two-sided.

mPFC: $n = 42, 42, 40, 40, 38, 37, 36, 35, 34, 33$ sessions for 1 to 10 dimension, Re: $n = 14$ for dimension 1, 2 and $n = 13$ for dimension 3 to 10. $P = 0.0412, 0.0397, 0.0231, 0.0215, 0.0126, 0.0095, 0.0053$ for dimension 3, 5, 6, 7, 8, 9, 10. Box plots showing median, 25%–75% percentile, minimum to maximum range. *$P < 0.05$, **$P < 0.01$, ***$P < 0.001$. Details of the statistical information are provided in Supplementary Data 1.

electrodes and socket were fixated by laying dental cement. Following the one-week post-surgery recovery period, mice were habituated to the Neurologger dummy device for three days. On the test day, the Neurologger was plugged in and the LFPs were recorded in the three-chamber box with empty cylinder cups for 5 min as the baseline. The sampling rate was 1600 Hz. Then the mouse was administered CNO and returned to a home cage. After ~35 min, LFPs were recorded again in the same three-chamber box. 5 min of LFP data were recorded for the habituation phase followed by a 10-min sociability phase and another 10-minute social novelty phase. The LFP data before the CNO injection were used as the baseline for normalization for the data analysis.

**Coherence analysis**

LFP data were analyzed using Matlab. To reduce noise, the 50 Hz line was removed by a notch filter and the signals were detrended. Power and coherence were analyzed using the Chronux toolbox[59]. The parameters were chosen as TW = 3, K = 5. Time-dependent coherence analysis of the LFP data focused on 3 s before and after the investigation started. The moving window length was 2 s and the step was 50 ms. The time-dependent coherence was normalized by dividing the pre-stimulus data across frequency. The mean coherence or power was calculated for each frequency band.

**In vivo electrophysiology using tetrodes**

Spiking activity was recorded in vivo in free moving animals using Plexon systems (OmniPlex Neural Recording Data Acquisition System and CinePlex Behavioral Research System). An overhead camera monitored the behavioral video simultaneously with the electrophysiological recordings. Spikes were sampled at 40 kHz with a bandpass filter from 300 Hz to 8 kHz and LFPs were sampled at 1 kHz with a bandpass filter from 0.7 Hz to 200 Hz. The multiple electrode array was customized made composing of 4 or 8 tetrodes. Each tetrode was made from 4 twisted insulation-coated platinum-iridium wires with a diameter of ~17 μm. Tetrodes were inserted into fine silica capillary tubes and attached to tunable screws on the customized PCB boards with a movable mechanism. Movable electrodes allowed recordings of multiple sessions in each animal. Electrode tips were plated with platinum to reduce the impedance in the range of 200–500 kΩ. The wires were inserted into the 32-channel Omnetics connector, and individual reference and ground electrodes were placed on the animal's skull. We included 22 and 15 mice for the mPFC and Re recording experiments, respectively. For the combined mPFC recording and Re DREADD inhibition experiment, we had 12 mice for both CNO and VEH groups. On average, three recording sessions were performed for each mouse.

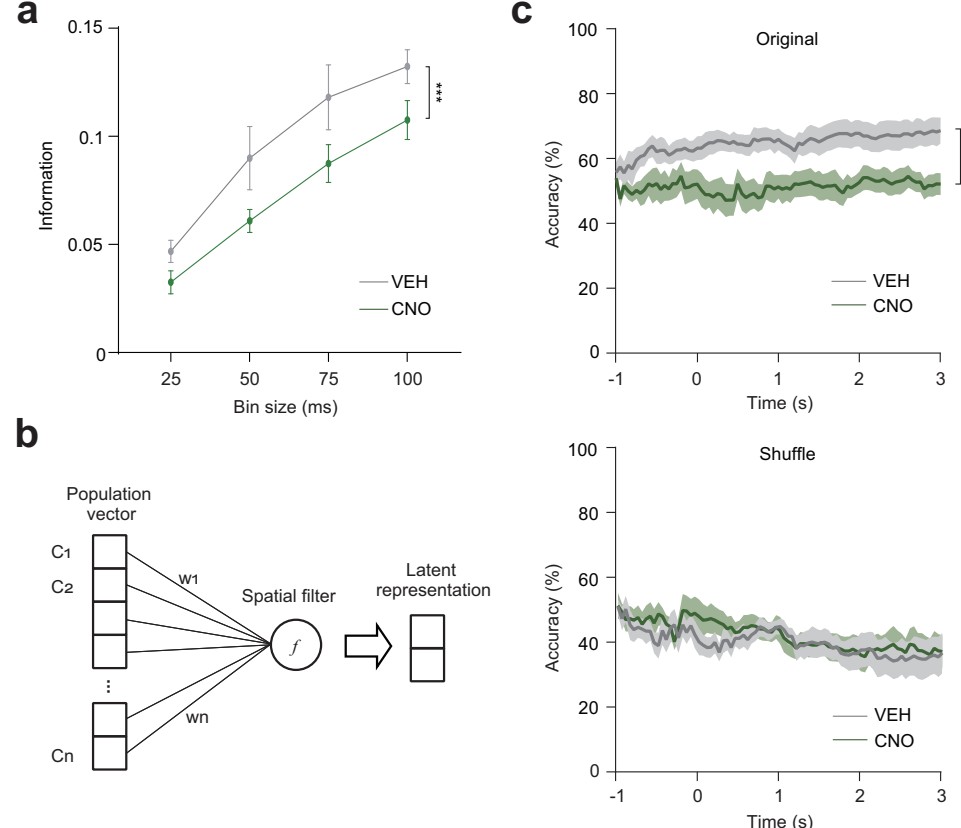

**Fig. 7 | Re supports mPFC social coding. a** mPFC information for S, E, S' and N in mice expressing hM4D in the Re under the treatment of CNO or VEH. VEH: $n = 23$ sessions, CNO: $n = 28$ sessions. Two-way ANOVA, treatment $F_{(3, 196)} = 28.27$, $P < 0.0001$. Data are mean ± s.e.m. **b** Schematic showing the spatial filters and the latent representations of population vectors. **c** Decoding accuracy in the classification of S and N using spatial filtering and single-trial decoding with original (top) and shuffle (bottom) data in mice expressing hM4D in the Re under the treatment of CNO or VEH. Shuffling was done by randomly pairing the stimulus labels and the neural data. VEH: $n = 9$ sessions, CNO: $n = 8$ sessions. Original accuracy, two-way ANOVA, treatment $F_{(80, 1200)} = 9.832$, $P = 0.0068$. Line and shaded areas are mean ± s.e.m. **P < 0.01. ***P < 0.001. Details of the statistical information are provided in Supplementary Data 1.

## Single-unit analysis

Spike sorting was performed offline using Offline Sorter 4.0 (Plexon systems). Principal component scores were used to cluster units and the L-ratio and isolation distance were calculated. A group of waveforms was considered to be generated from one single unit if it was distinct from other clusters. In the principal component space, the separation of the clusters was determined by manual inspection. The dead time was set to 1 ms for the refractory period. Cross-channel validation was done using cross-correlation histograms. If the peak activity of any two neurons coincided, only one of them was considered. Only well-isolated spikes were included for the analysis. To examine the social behavior when the mice approached and retreated from the stimuli, we manually analyzed the social interaction behavior by visually examining the behavioral videos on a frame-by-frame basis. In each bout, when the animals approached the cup, performed active investigation and retreated from it, we labeled the frames corresponding to the start and the finish of the investigation. This video analysis was also aided by a deep learning tool[60] to locate the investigation frames more quickly.

Single-unit spike data were converted to point processes at 1 ms with 0/1, and the firing rate was calculated using a 100 ms bin. Neurons with a mean firing rate of less than 0.5 Hz were not used. We included the epochs with an investigation duration longer than 3 s. To evaluate the response of a neuron, a non-parametric signed-rank test was used to compare the firing rates of the 3 s data before and after the onset of investigation across all trials. A $P$ value less than 0.05 was used as the criterion for being responsive. To examine the response profiles for each stimulus, firing rate changes within each stimulus were averaged across trials. Then the firing rate was converted to a z-score by normalizing the baseline. When the data were aligned at the beginning of the investigation, we subtracted the pre-stimulus period as the baseline. When the data were aligned at the end of the investigation, we subtracted the during-stimulus period as the baseline. When neurons contained no baseline firing for certain stimuli, we estimated the baseline by resampling the baseline periods of all neurons in the recording session. The procedure was performed 1000 times, and the mean was used as the baseline.

The selectivity index was calculated as

$$SI = \frac{\mu_1 - \mu_2}{\mu_1 + \mu_2} \tag{1}$$

$\mu_1$ and $\mu_2$ are the mean firing rates of the two stimuli[11].

## Population vector analysis

To calculate the population response, the population vector was calculated from the firing rates of all cells in the session. Only sessions in which at least eight cells were recorded simultaneously were used. We calculated the Mahalanobis distance between pre-stimulus (3 s before the onset of investigation) and during-stimulus (3 s after the onset of investigation). The calculation formula was as follows:

$$M_{sj} = \sqrt{\left(\mathbf{D}_{sj} - \mathbf{P}_{sj}\right)^T \mathbf{C}_{sj}^{-1} \left(\mathbf{D}_{sj} - \mathbf{P}_{sj}\right)} \tag{2}$$

was the Mahalanobis distance between pre-stimulus and during-stimulus population vectors in the $j$-th trial of stimulus $s$ from S, E, S' and N. $\mathbf{D}_{sj}$ was the during-stimulus population vector. $\mathbf{P}_{sj}$ was the pre-stimulus population vector. $\mathbf{C}_{sj}$ was the covariance matrix. We used the maximum Mahalanobis distance of all trials for each stimulus in each session.

To evaluate the responsiveness of the population vector, we used a bootstrap method to shuffle the trials containing the population vectors across the stimuli. The procedure was repeated for 1000 times in the same recording session and a shuffled Mahalanobis distance was calculated.

## Information analysis

Information was calculated based on the Shannon entropy[39]. Entropy quantifies the uncertainty about which stimulus is presented and the average amount of information gained with each stimulus. We calculated the entropy of binned population vectors related to a given stimulus. In a recording session for a given stimulus $s \in$ S,E,S' and N, the simultaneously recorded neurons constituted the population vector. We used time bins of size 20, 50, 75 and 100 ms to count the spikes. For an investigation bout of 3 s, this gave 150, 60, 40 and 30 binned population vectors. The bin vectors were combined across all the bouts. The spike counts in each bin were discretized into three levels. The calculation formula was as follows

$$I = H(\mathbf{f}) - H(\mathbf{f}|s) \tag{3}$$

$$H(\mathbf{f}) = -\sum_{\mathbf{f}} p(\mathbf{f}) \log_2 p(\mathbf{f}) \tag{4}$$

$$H(\mathbf{f}|s) = -\sum_{s} p(s) \sum_{\mathbf{f}} p(\mathbf{f}|s) \log_2 p(\mathbf{f}|s) \tag{5}$$

$I$ was the information for the population vectors. $p(s)$ was the occurrence of the stimulus and was calculated as the bout number of stimulus $s$ divided by the number of all bouts. $\mathbf{f}$ was the binned population vector for a given stimulus. We used 8 cells in the population vectors to calculate the information. Recording sessions with fewer cells were not used and when the sessions had more cells, we randomly selected eight cells. Information estimation may have bias, and we corrected $I$ using a bias correction method[39,61].

## Population decoding

Population decoding was performed within each session based on binned population vectors from the during-stimulus period. The time bin was 300 ms. Before decoding, the baseline was corrected for each trial by subtracting the mean firing of pre-stimulus data. For an investigation bout of 3 s, this gave 10 binned population vectors. The binned population vectors were combined across all bouts. Baseline-corrected population vectors were treated as the features and the corresponding stimulus types were treated as the true labels. A naïve Bayesian decoder was trained[62]. Under the assumption that firing of $n$ neurons $f_1, f_2, \cdots, f_n$ in a population vector for a given stimulus was independent of each other, the likelihood of occurrence of a stimulus could be calculated, following the Bayesian rule, as the posterior probability

$$p(s|f_1, f_2, \cdots, f_n) \propto p(s) \prod_{c=1}^{n} g(f_c|s) \tag{6}$$

The prior probability $p(s)$ was the occurrence of stimulus $s$. The firing distribution of neuron $c$ given stimulus $s$, $g(f_c|s)$, could be modeled as a Gaussian distribution, i.e., $g(f_c|s) \sim N(\mu_{c|s}, \sigma_{c|s})$, where the mean and the standard deviation $\mu_{c|s}$ and $\sigma_{c|s}$ were obtained by the maximum likelihood estimation performed on training samples.

The inferred stimulus $\hat{s}$ was the stimulus with the maximum posterior probability

$$\hat{s} = \underset{s}{\mathrm{argmax}} \, p(s|f_1, f_2, \cdots, f_n) \tag{7}$$

The leave-one-out cross-validation was applied to evaluate each session's decoding performance. This procedure predicted the label of one sample each time, using the model trained on all other samples, and was repeated until all samples had been predicted once. The decoding accuracy of each stimulus was calculated as the percentage of correctly predicted labels. Because the bout number for each stimulus was different, balanced decoding accuracy, obtained by taking the average of decoding accuracies of each stimulus, was calculated.

We used sessions with more than 10 available cells when we examined the decoding accuracy across vector dimensions of different cell number. The same recording sessions were used and number of cells corresponded to the vector dimension. We calculated the variances for the cells and ranked them. For each dimension $n$, the first $n$ cells with the largest variances were used.

## Spatial filter

The spatial filter projects the population vectors to a low-dimensional space by the latent representations[63]. It uses supervised learning to separate the latent representations from the two categories maximally so that the event-related response is isolated and the noise is suppressed. We used binned population vectors $\mathbf{X}_{s,n}$ in each trial $n$ of the stimulus $s$ to perform the filtering between two stimuli. The population vector came from the sets $\{\mathbf{X}_{s,n} \in \mathbf{R}^{m \times t} | s = 1,2; n = 1,2, \cdots, n_s\}$ with $m$ being the cell number and $t$ the time bin. $n_s$ was the number of trials for stimulus $s$. We calculated the center for the population vector sets in each stimulus as

$$\overline{\mathbf{X}_s} = \frac{1}{n_s} \sum_{i=1}^{n_s} \mathbf{X}_{s,i}, s = 1,2. \tag{8}$$

The distance between the two centers of two stimuli was $\mathbf{A} = (\overline{\mathbf{X}_1} - \overline{\mathbf{X}_2})(\overline{\mathbf{X}_1} - \overline{\mathbf{X}_2})^{\mathsf{T}}$. The distance within the stimulus was

$$\mathbf{B} = \frac{1}{2} \sum_{s=1}^{2} \frac{1}{n_s} \sum_{i=1}^{n_s} (\mathbf{X}_{s,i} - \overline{\mathbf{X}_s})(\mathbf{X}_{s,i} - \overline{\mathbf{X}_s})^{\mathsf{T}} \tag{9}$$

The spatial filter relied on the generalized eigen decomposition $\mathbf{Au} = \lambda \mathbf{Bu}$ and an eigenvector $\mathbf{u}$ was found to maximize the between-stimulus distance and minimize the within-stimulus distance. $\lambda$ was the eigenvalue. When $\mathbf{B}$ was not positive definite, we calculated $\hat{\mathbf{B}} = (1 - \epsilon)\mathbf{B} + \epsilon\mathbf{I}$ as a substitute with $\epsilon = 0.01$. The latent space $\mathbf{P} \in \mathbf{R}^{m \times p}$ used the first $p$ eigenvectors sorted by corresponding eigenvalues in descending order to transform $\mathbf{X}_{s,n}$. The final projected latent representation is $\hat{\mathbf{X}} = \mathbf{P}^{\mathsf{T}} \mathbf{X}_{s,n}$.

## Single-trial decoding

We performed the single-trial decoding analysis between two stimuli based on a time-dependent cumulative probability model using the Bayes rule[64]

$$\hat{s} = \underset{s}{\mathrm{argmax}} \, \log p(s) + \sum_{c} \sum_{j} \log g\left(f_c(t_j)|s\right) \tag{10}$$

$f_c(t_j)$ was the binned population vector with $t_j$ as the $j$-th time bin. $\hat{s}$ was the inferred stimulus. Unlike the naïve Bayesian decoder using the population vectors described above, we treated the firing rate at the current time bin as independent random variables and calculated a posterior as the cumulative distribution across time. Then we obtained the likelihood for each stimulus type.

Decoding directly using population vectors between two stimuli S and N can be difficult, we therefore performed spatial filtering on the population vectors first. Then the latent representation went into the above likelihood estimation for time-dependent decoding. We used sessions with at least 8cells and at least eight trials for stimuli S and N. The population vector dimension was chosen as 8. The time bin was 50 ms. 8 trials were selected for each stimulus. Leave-one-out cross-validation was performed. One of 8 trials was left as the test and the remaining trials were used for training. To test the decoding performance, we shuffled the stimulus labels for the data across the trials as a control procedure.

### Statistical analysis
Statistical analysis was performed using Prism GraphPad 7. The details of the statistical analysis are provided in Supplementary Data 1.

### Reporting summary
Further information on research design is available in the Nature Portfolio Reporting Summary linked to this article.

## Data availability
The authors declare that all source data supporting the claims in this study are provided as a Source Data file. Other data that support the findings of this study are available upon request. Source data are provided with this paper.

## Code availability
The Matlab code is available on GitHub at https://github.com/YangZhanLab/prefrontal-thalamic-social-recognition.

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

## Acknowledgements

We thank Hongjian Li, Wanying Li, Meng Yang, Yao Wang and Dengyun Ge for technical assistance. We thank Tyler Brown and Guoping Feng for critical reading of the manuscript. This work was supported by National Natural Science Foundation of China grant No. 31671101 (Y.Z.), No. 32070985 (Y.Z.), Guangdong Provincial Key Laboratory of Brain Connectome and Behavior 2023B1212060055, Shenzhen Key Laboratory of Translational Research for Brain Diseases No. ZDSYS20200832815480001 (Y.Z.).

## Author contributions

Z.C., Y.H., X.W., Y.T. performed optogenetic experiments. Y.H. collected and analyzed LFP data and performed chemogenetic experiments. S.X. and Z.C. performed in vivo single unit electrophysiology. Z.C. and Y.Z. performed data analysis for single-unit spike data. B.S. and Y.Z. performed information anlaysis. Z.M. performed decoding analysis. Z.C., Y.H. and X.W. performed statistical analysis on behavioral data. A.L.V. provided the electrophysiology hardware. Y. Z. and B.S. performed research design. Y.Z. conceived the project. Y.Z. wrote the paper with inputs from Z.C., Y.H., Z.M. and B.S.

## Competing interests

The authors declare no competing interests.
