## [Peer Review File · Nature Communications]

REVIEWER COMMENTS

Reviewer #1 (Remarks to the Author):

This review was prepared by Kay M. Tye. Towards an effort to increase transparency and accountability in the peer review process, and to expedite publication of scientific research, I no longer participate in anonymous peer review (Effective Jan 1, 2018). This review is provided with the understanding that my name will be made available to the authors and to the other reviewers.

Overall, the authors have addressed the main concerns satisfactorily from the prior round of review, and I support publication of this manuscript in Nature Communications! I have just a few easy and optional suggestions for improving the clarity and presentation of their data, and I do not necessarily need to see the manuscript for re-review.

Minor Comments:

For Figure 1: please plot the hierarchical clustering such that each row is the same neuron, because as it is, they apparently just sorted all the neurons recorded for each event, preventing the reader from being able to observe the overall response profile. (If there was a cell that selectively encoded social proximity, I would not be able to figure out which row corresponds to the same neuron as its currently plotted). I also don't understand the point of the analysis in 1B – is this the average of all responses to all stimuli? As it currently stands, I don't understand the takeaway of this plot.

Figure 2 G and I, simple statistical fix: should be plotted all together to look for an interaction between all the groups, because the key statistical comparison should be between DREADDs-CNO and mCherry-CNO, not for vehicle and CNO given the known off-target effects of CNO.

Not sure it makes sense to make population averages of behavioral events that are all different – every precise motor pattern is going to be slightly different and there will be huge order effects (as there are for all neural responses to social stimuli, as they are rapidly habituating). Have the authors considered applying pseudopopulations to their decoding analyses?

I was happy to see that they looked at decoding accuracy in PFC neurons to make a claim about social recognition (Fig 6) , and to see that this was affected by Re-inhibition. I commend the authors for their efforts to revise and improve their manuscript.

Reviewer #3 (Remarks to the Author):

The authors completed an impressive multidisciplinary study of social behaviors mediated by a pathway from the medial prefrontal cortex to the thalamic nucleus reuniens. Overall, most of the previous concerns have been addressed. Additional behavioral testing was conducted using larger group Ns. Minor wording changes were made to effectively modify previous misstatement and overstatements.

The authors responded to the key issue of general exploratory behavior fairly well, by displaying the absence of group differences in number in entries between chambers during the habituation phase.

Please note that representative tracks and heat maps are illustrative only. Graphs displaying means, standard errors of means, and statistical analyses of raw data are necessary, and should be given prominence as the primary data presentation in Figures 3 and 5. Further, speed and acceleration are general measures of motor ability, and do not directly reflect interest in exploring the environment. The authors are encouraged to include a specific control assay for general exploratory locomotion in future studies.

Similarly, future experiments using the 3-chambered social approach apparatus should employ habituation sessions in which all 3 chambers are completely empty, rather than allowing familiarization to the wire cup during habituation. Although the authors wrote long justifications for their procedure, in fact the comparison during the social approach phase should be between a novel mouse and the completely novel wire cup object.

Note that novel object recognition is a separate assay without a social component, and therefore does not serve as a control for 3-chambered social recognition.

While an odor recognition test was conducted, it is not stated whether the different odors were social or non-social. The goal should be to evaluate whether the subject mouse can differentiate odors from two

different mice, which is a sensory ability essential for preference for social novelty. However, since a separate social memory test was conducted that employed familiar versus nonfamiliar partners, it appears that both groups had adequate social odor recognition abilities.

Response to reviewers' comments

The reviewers' comments are marked in blue color. We highlighted the changed text contents with blue font in the main manuscript.

Reviewer #1 (Remarks to the Author):

****This review was prepared by Kay M. Tye. Towards an effort to increase transparency and accountability in the peer review process, and to expedite publication of scientific research, I no longer participate in anonymous peer review (Effective Jan 1, 2018). This review is provided with the understanding that my name will be made available to the authors and to the other reviewers.****

Overall, the authors have addressed the main concerns satisfactorily from the prior round of review, and I support publication of this manuscript in Nature Communications! I have just a few easy and optional suggestions for improving the clarity and presentation of their data, and I do not necessarily need to see the manuscript for re-review.

Minor Comments:

For Figure 1: please plot the hierarchical clustering such that each row is the same neuron, because as it is, they apparently just sorted all the neurons recorded for each event, preventing the reader from being able to observe the overall response profile. (If there was a cell that selectively encoded social proximity, I would not be able to figure out which row corresponds to the same neuron as its currently plotted). I also don't understand the point of the analysis in 1B – is this the average of all responses to all stimuli? As it currently stands, I don't understand the takeaway of this plot.

Response: We thank the reviewer for raising this point. As the reviewer pointed out, we showed the enhanced responses of individual neurons to examine the response profiles and to compare the profiles across different stimulus categories. However, the figure displays focused on the peak response but they did not disclose the differences across the stimuli. Since we selected responsive neurons using all the investigation bouts across all four stimuli, we now rearranged the neuron orders according to the average magnitude of the four stimuli. The new data figure shows the differences between stimuli. We have updated Figure 1d, e and Figure 3d, e.

Figure 2 G and I, simple statistical fix: should be plotted all together to look for an interaction between all the groups, because the key statistical comparison should be between DREADDs-CNO and mCherry-CNO, not for vehicle and CNO given the known off-target effects of CNO.

Response: We thank the reviewer for this point. We have now combined the four groups and performed a two-way ANOVA on the coherence. We found a drug-by-group interaction for gamma-band coherence. The results are shown in Figure 2i.

Not sure it makes sense to make population averages of behavioral events that are all different – every precise motor pattern is going to be slightly different and there will be huge order effects (as

there are for all neural responses to social stimuli, as they are rapidly habituating). Have the authors considered applying pseudopopulations to their decoding analyses?

Response: We performed a single-trial decoding analysis to see whether, during the investigation time, the firing response could represent different stimuli. We did not observe motor differences across the four stimuli (Figure S1b, c), indicating that when investigating the different stimuli, the mice moved the body with comparable extents. As the reviewer pointed out, motor activity during the investigation may correlate with the firing changes, this may be investigated through a multiple regression model in which speed can be treated as a contributing variable. However, in the current study, we performed a spatial filtering on the firing response aiming to remove the common influences driven by the noise. As the review pointed out, we did not generate pseudocopulation such as introducing jitters in the spike time, but we shuffled the trial labels to generate the control datasets and tested the decoding ability of our model on the shuffled data.

I was happy to see that they looked at decoding accuracy in PFC neurons to make a claim about social recognition (Fig 6) , and to see that this was affected by Re-inhibition. I commend the authors for their efforts to revise and improve their manuscript.

Response: We thank the reviewer for the comment.

Reviewer #3 (Remarks to the Author):

The authors completed an impressive multidisciplinary study of social behaviors mediated by a pathway from the medial prefrontal cortex to the thalamic nucleus reuniens. Overall, most of the previous concerns have been addressed. Additional behavioral testing was conducted using larger group Ns. Minor wording changes were made to effectively modify previous misstatement and overstatements.

The authors responded to the key issue of general exploratory behavior fairly well, by displaying the absence of group differences in number in entries between chambers during the habituation phase.

Please note that representative tracks and heat maps are illustrative only. Graphs displaying means, standard errors of means, and statistical analyses of raw data are necessary, and should be given prominence as the primary data presentation in Figures 3 and 5. Further, speed and acceleration are general measures of motor ability, and do not directly reflect interest in exploring the environment. The authors are encouraged to include a specific control assay for general exploratory locomotion in future studies.

Response: Since the total time in the chamber contained other behavioral processes and did not directly reflect the measurements of social investigation, we now used the direct interaction time to measure the actual sniffing time towards the social and non-social targets. In addition, we also calculated the preference index between the two stimuli for both sociability and social novelty

phases. We also present the heat maps showing the cumulative time as examples. This way of data presentation has been extensively used in other studies employing the three-chamber social interaction test ^{1, 2, 3}. The updated figures are now shown in Figure 4b, Figure 5, and Figure S8.

Similarly, future experiments using the 3-chambered social approach apparatus should employ habituation sessions in which all 3 chambers are completely empty, rather than allowing familiarization to the wire cup during habituation. Although the authors wrote long justifications for their procedure, in fact the comparison during the social approach phase should be between a novel mouse and the completely novel wire cup object.

Response: We thank the reviewer for the comment. There are procedural differences regarding the three-chamber test. For instance, in this nice paper which described the procedure in detail ², the authors used a separate habituation session during which the mice explored the empty environment. Then the procedure continued with another exploratory session in which the cups were presented and filled with paper balls. There are also papers using the habituation paradigm in which the mice explored the environment with the placement of empty cups ^{3, 4, 5}. We think that it may be a good point regarding the habituation session in which nothing is presented in the environment because the empty cup would be completely novel during the subsequent sociability phase. In the future, we would like to adopt this procedure. We will also consider using a fake toy mouse inside the empty cup during the sociability phase in future experiments.

Note that novel object recognition is a separate assay without a social component, and therefore does not serve as a control for 3-chambered social recognition.

Response: We agree with the reviewer that we employed the novel object recognition test to examine the ability to recognize novel objects when the Re was inhibited. We changed the text and removed “as another control experiment”. We thank the reviewer for pointing this out.

While an odor recognition test was conducted, it is not stated whether the different odors were social or non-social. The goal should be to evaluate whether the subject mouse can differentiate odors from two different mice, which is a sensory ability essential for preference for social novelty. However, since a separate social memory test was conducted that employed familiar versus nonfamiliar partners, it appears that both groups had adequate social odor recognition abilities.

Response: We thank the reviewer for raising the point regarding the odor being social or non-social. Indeed, in the methods section, we stated that “A pinch of sawdust bedding containing the social odor from the same home cage where the test mouse was housed was randomly put into one of the cups.” Therefore, we used social odors. The first social odor came from a home cage and the second social odor came from another home cage. However, as the reviewer pointed out, in the main text we did not mention whether it was a social or non-social odor. In the updated manuscript, as the reviewer suggested, we have now specifically mentioned that the odor came from the soiled bedding of a home cage.

References

1. Qin, L. *et al.* Social deficits in Shank3-deficient mouse models of autism are rescued by histone deacetylase (HDAC) inhibition. *Nat Neurosci* **21**, 564-575 (2018).
2. Rein, B., Ma, K. & Yan, Z. A standardized social preference protocol for measuring social deficits in mouse models of autism. *Nat Protoc* **15**, 3464-3477 (2020).
3. Lopez-Rojas, J., de Solis, C.A., Leroy, F., Kandel, E.R. & Siegelbaum, S.A. A direct lateral entorhinal cortex to hippocampal CA2 circuit conveys social information required for social memory. *Neuron* **110**, 1559-1572.e1554 (2022).
4. Rapanelli, M. *et al.* Behavioral, circuitry, and molecular aberrations by region-specific deficiency of the high-risk autism gene Cul3. *Mol Psychiatry* **26**, 1491-1504 (2021).
5. Wu, X., Morishita, W., Beier, K.T., Heifets, B.D. & Malenka, R.C. 5-HT modulation of a medial septal circuit tunes social memory stability. *Nature* **599**, 96-101 (2021).

REVIEWERS' COMMENTS

Reviewer #3 (Remarks to the Author):

The authors have effectively addressed the previous concerns.

Response to reviewers' comments

The reviewers' comments are marked in blue color. We highlighted the changed text contents with blue font in the main manuscript.

Reviewer #3 (Remarks to the Author):

The authors have effectively addressed the previous concerns.

Response: We thank the reviewer for the comment.